# Characterization of Two-Component System CitB Family in *Salmonella* Pullorum

**DOI:** 10.3390/ijms231710201

**Published:** 2022-09-05

**Authors:** Xiamei Kang, Xiao Zhou, Yanting Tang, Zhijie Jiang, Jiaqi Chen, Muhammad Mohsin, Min Yue

**Affiliations:** 1Institute of Preventive Veterinary Sciences, Department of Veterinary Medicine, Zhejiang University College of Animal Sciences, Hangzhou 310058, China; 2Hainan Institute, Zhejiang University, Sanya 572025, China; 3Zhejiang Provincial Key Laboratory, Preventive Veterinary Medicine, Hangzhou 310058, China

**Keywords:** *Salmonella* Pullorum, CitB family, Two-component system, Pathogenesis

## Abstract

*Salmonella enterica,* serovar Gallinarum, biovar Pullorum, is an avian-specific pathogen which has caused considerable economic losses to the poultry industry worldwide. Two-component systems (TCSs) play an essential role in obtaining nutrients, detecting the presence of neighboring bacteria and regulating the expression of virulence factors. The genome analysis of *S*. Pullorum strain S06004 suggesting the carriage of 22 pairs of TCSs, which belong to five families named CitB, OmpR, NarL, Chemotaxis and LuxR. In the CitB family, three pairs of TCSs, namely CitA-CitB, DcuS-DcuR and DpiB-DpiA, remain unaddressed in *S*. Pullorum. To systematically investigate the function of the CitB family in *S*. Pullorum, four mutants, Δ*citAB* (abbreviated as Δ*cit*), Δ*dcuSR* (Δ*dcu*), Δ*dpiBA* (Δ*dpi*) and Δ*citAB*Δ*dcuSR*Δ*dpiBA* (Δ3), were made using the CRISPR/Cas9 system. The results demonstrated that the CitB family did not affect the growth of bacteria, the results of biochemical tests, invasion and proliferation in chicken macrophage HD-11 cells and the expression of fimbrial protein. But the mutants showed thicker biofilm formation, higher resistance to antimicrobial agents, enhanced tolerance to inhibition by egg albumen and increased virulence in chicken embryos. Moreover, the deletion of Dpi TCS was detrimental to survival after exposure to hyperosmotic and oxidative environments, as well as the long-term colonization of the small intestine of chickens. Collectively, we provided new knowledge regarding the possible role of the CitB family involved in the pathogenic processes of *S*. Pullorum.

## 1. Introduction

*Salmonella enterica,* serovar Gallinarum, biovar Pullorum (*S.* Pullorum), a causative agent of Pullorum disease in poultry [1], is among the most important pathogens which have resulted in substantial economic losses to the poultry industry worldwide [2,3,4]. Pullorum disease causes severe septicemia, mainly in young birds, with clinical symptoms such as loss of weight, decreased laying, diarrhoea, lesions and disorder of the reproductive system [2,5,6,7,8]. Our previous meta-analysis showed that *S*. Pullorum is still very common in many regions, especially in developing countries [9], although it has been the target of disease eradication schemes in most industrialized countries [2,10,11,12].

Two-component regulatory systems (TCSs), which are employed to sense and respond to various environmental stresses, are critically important in regulating the virulence determinants of many bacterial pathogens, consisting of sensor kinases and response regulators [13,14,15]. At present, there are in-depth studies on the TCSs in *Salmonella*, including the PhoP-PhoQ system mainly controlling virulence and Mg^2+^ homeostasis [16,17], the PmrA-PmrB system conferring resistance to the peptide antibiotic polymyxin B [18,19,20], and the RcsB-RcsC system modulating the expression of invasion proteins and flagellin [21,22]. Twenty-two pairs of TCSs belonging to five families have been identified through the genome analysis of *S*. Pullorum strain S06004, which are named OmpR (eleven pairs), Chemotaxis (four pairs), CitB (three pairs), NarL (three pairs) and LuxR (one pair). Currently, studies on the pathobiological function of the CitB family in *S*. Pullorum remain unknown.

The CitB family, containing three pairs of TCSs, namely the CitA-CitB, DcuS-DcuR and DpiB-DpiA systems, is a widespread signal transduction family in gram-negative bacteria. Previous studies have confirmed that CitA-CitB TCS is essential for the induction of the citrate fermentation genes [23,24], DpiB-DpiA TCS, which is an orthologue of the CitA-CitB system that regulates citrate uptake and utilization [25], can induce SOS in response to certain β-lactams [26] and DcuS-DcuR TCS stimulates the expression of genes such as *dcuB* involved in C4-dicarboxylate metabolism [27]. In this study, we constructed four mutant strains of the CitB family, Δ*citAB* (abbreviated as Δ*cit*), Δ*dcuSR* (Δ*dcu*), Δ*dpiBA* (Δ*dpi*) and Δ*citAB*Δ*dcuSR*Δ*dpiBA* (Δ3), to investigate the functional differences among the three TCSs and to investigate the possible role of pathogenic mechanisms for *S.* Pullorum, and the gained knowledge will provide additional evidence of the TCSs of the CitB family.

## 2. Results

### 2.1. Deletion of TCSs of the CitB Family Does Not Affect Growth and the Results of Biochemical Tests

To investigate the independent roles of the three pairs of TCSs of the CitB family in *S.* Pullorum, mutant strains Δ*cit*, Δ*dcu*, Δ*dpi* and Δ3 were constructed using the CRISPR/Cas9 editing system [28]. The correct mutants were confirmed by a polymerase chain reaction (PCR) (Appendix A) and Sanger sequencing. There were no significant differences in growth curves between wild-type (WT) and mutant strains under LB (Lennox broth) shaking, LB static culture and M9 minimal medium static culture (Figure 1A–C). The growth of all five strains under aerobic conditions was much better than that under anaerobic conditions, which is consistent with the characteristic that *S.* Pullorum is facultative anaerobic. Among the fifteen kinds of microtubes tested for biochemical tests (Figure 1D), all five strains showed positive results for seven substances (glucose (gas production), L-arabinose, L-rhamnose, trehalose, mannitol, ornithine decarboxylase and 3% catalase), but the other eight were negative (oxidase, arginine dihydrolase, xylose, tartrate, mucate, D-sorbitol, dulcitol and phenylalanine). These results indicated that the CitB family did not affect growth and the results of the biochemical tests.

### 2.2. Deletion of TCSs of the CitB Family Increases the Biofilm Formation and the Resistance to Aminoglycosides Drugs under Anaerobic Conditions

To study whether the genes of the CitB family regulate biofilm formation, the ability of biofilm formation was examined under different conditions. The results showed that all strains were weak biofilm producers under aerobic conditions (Figure 2A,C). However, the biofilm formation of the mutant strains was significantly enhanced under anaerobic conditions compared to the WT strain whether there were bile salts or not (Figure 2B,D). Of the three bile salts tested, pig bile salt was the most effective in stimulating biofilm formation (Figure 2C,D). Furthermore, when grown only in LB broth without bile salts, the ability of biofilm formation at 20 °C and 28 °C was more robust than that at 37 °C and 42 °C (Figure 2B).

Minimal inhibitory concentration (MIC) assays were conducted using the broth microdilution method based on the Clinical and Laboratory Standards Institute (CLSI) guidelines [29]. Our findings demonstrated that the resistance profiles of all of the strains were the same under aerobic conditions except for ceftiofur (CF), to which the deletion mutants were more resistant than WT (Figure 2E). Interestingly, the deletion of the CitB family increased the resistance toward CF and aminoglycosides such as gentamicin (GEN) and kanamycin (KAN) under anaerobic conditions. In addition, The MIC values of aminoglycosides and quinolones were both lower under aerobic conditions than those under anaerobic conditions, while the opposite was observed for tetracyclines, polymyxins and sulfonamides. Taken together, the mutants of the CitB family showed thicker biofilm and higher resistance to antimicrobials under anaerobic conditions.

### 2.3. The Dpi TCS Contributes to the Survival of S. Pullorum in Hyperosmotic and Oxidative Environments

Since TCSs are responsible for sensing and responding to various environmental stresses, we evaluated the survival ability of the WT and mutant strains towards different environmental stresses. As seen in Figure 3, the strains showed more tolerance to hyperosmotic and oxidative stresses, verifying that *S.* Pullorum, as a facultative intracellular pathogen, was adapted to the intracellular environment, in which the host cells produced some antimicrobial molecules including reactive oxygen species [30,31]. More importantly, the Dpi TCS was essential for the persistence in the hyperosmotic and oxidative environments as its depletion resulted in a significantly decreased survival ratio compared to WT after stresses. Additionally, the CitB family did not change the tolerance to acid stress or heat stress.

### 2.4. Deletion of TCSs of the CitB Family Increases the Growth in Egg Albumen and Virulence to Chicken Embryo

Previous studies have elaborated that stress-induced survival mechanisms may enable *S*. Pullorum to cope with the antimicrobial compounds such as lysozyme and antimicrobial peptides in egg albumen [7]. Consequently, we tested the ability of all five strains to grow in egg albumen. After incubation at 37 °C for 24 h with the same inoculum, four mutants exhibited a considerable increase in contrast to WT, even though only Dcu TCS had a statistical difference (Figure 4A). Further, the virulence of the WT and mutant strains was compared to an in vivo chicken embryo model. The results showed that the mutants led to higher mortality than WT (Figure 4B). However, there was no significant difference in the bacterial loads in the liver and spleen between WT and mutants (Figure 4C,D).

### 2.5. Deletion of Dpi TCS Is Detrimental to Long-Term Colonization of the Small Intestine but Does Not Affect the Expression of the Fimbrial Protein

The virulence and colonization were not affected by the Cit and Dcu TCSs of the CitB family. However, the bacterial loads of Δ*dpi* mutant in the small intestine significantly decreased 14 days post-infection, suggesting that the Dpi TCS may contribute to long-term colonization (Figure 5A–D).

Fimbriae, one of the virulence factors in *Salmonella*, are capable of mediating adherence and colonization to tissues such as intestinal mucosa [32,33]. To further explore whether the CitB family, especially the Dpi TCS, is associated with the expression of a fimbrial adhesin, the polyclonal antibodies of major subunits of six sets of the fimbrial system in *S*. Pullorum (BcfA, FimA, LpfA, SafA, StdA and SthA) were prepared [32,34,35]. Western blotting analysis showed that the protein expression levels of the mutants were the same as the WT except for SafA and SthA, which had no target protein bands due to their low expression (Figure 5E). Hence, the TCSs of the CitB family did not affect the expression of the fimbrial protein.

## 3. Discussion

*S.* Pullorum, causing Pullorum disease in poultry, is an important pathogen in great need of control and disease eradication [3,5,9]. Nevertheless, the mechanism of how *S.* Pullorum targets and continuously colonizes the host’s organs remains obscure. It has been stated that bacterial TCSs are related to virulence genes such as *Salmonella* Pathogenicity Island-1 (SPI-1) genes [36]. The PhoP-PhoQ and CpxR-CpxA TCSs were reported to attenuate the expression of the virulence genes of *Salmonella* [37,38]. However, the role of TCSs of the CitB family in *S.* Pullorum has not yet been studied. In this study, we investigated the contribution of the CitB family to *S*. Pullorum pathogenicity.

Our results revealed that the CitB family appeared to have little effect on the *S*. Pullorum pathogenicity, which was previously highlighted as inducing the citrate fermentation genes and C4-dicarboxylate metabolism genes in other species of pathogens [23,24,27]. Despite harboring genetic alterations, the strains showed an equal uptake and proliferation rate when infecting HD-11 cells, exhibited similar biochemical test results and showed identical growth rates in LB broth and M9 minimal medium. However, they failed to grow when the glucose in the M9 medium was replaced with an equal concentration of citrate, indicating that the WT and mutants were not able to utilize the citrate (Appendix A) as previously reported [39]. These findings suggest that the CitB family genes are not required for the growth and substance usage of *S.* Pullorum. As previously reported, the CitB family genes are essential for citrate utilization in *Klebsiella pneumoniae* [23] and *Vibrio cholerae* [24]. In addition, Yamamoto et al. found that the genes related to citrate fermentation were mainly activated by the CitAB master regulatory system in *E. coli* [40]. However, our results showed a different situation in *S.* Pullorum. There are two possible explanations. One is that the CitB family may behave differently among different species. The other is that *S.* Pullorum is host-specific, and some gene functions have been lost during evolution [41]. The CitB family genes may have developed into pseudogenes to lessen the living burden and strengthen the survival capacity during within-host adaptation in later evolution [5,42].

In spite of the overall perspective above, the TCSs of the CitB family negatively controlled biofilm formation in *S.* Pullorum from the stimulated biofilm in four deletion mutants (Figure 2B,D), in concert with the role of the *pagC* and *rstA* genes in forming biofilm both activated by the PhoP-PhoQ TCS, which may indicate some relationship between the CitB family TCSs and PhoP-PhoQ TCS [43,44]. However, in this study, we did not explore this further. Moreover, the antimicrobial susceptibility test and growth in egg albumen exhibited that resistance to unfavourable environments was probably negatively regulated by the CitB family. Under the premise that biofilms aid pathogens in facilitating survival in unfavourable environments [43,45], it is easy to assume that a lack of the CitB family may lead to the upregulation of operons involved in biofilm formation. In this instance, the fact that the mutants were more virulent to chicken embryos could also make sense because they could more resist the adverse environment presented in chick embryos due to biofilm formation after challenge [7]. 

The TCSs of pathogens are responsible for sensing and responding to environmental changes to make adaption quickly [46]. The stress assays showed that the CitB family was necessary for survival in a hyperosmotic or oxidative environment as expected, especially Dpi TCS, which also affected the colonization in chickens orally infected. Altogether, all of this offered new information on the impact of Dpi TCS, depicted as formerly controlling citrate uptake and utilization [47]. Interestingly, the strains grew after hyperosmotic and oxidative stresses, which was different from what we expected. One possible explanation is that *S.* Pullorum, as a facultative intracellular pathogen, is adapted to the intracellular environment [30,31]. Therefore, environmental stimuli related to the host trigger the protective mechanism. Multi-omics analysis technology can be used to explore the relevant pathways and mechanisms in future research. 

It has been confirmed that fimbriae are responsible for adhering and colonizing tissues such as intestinal mucosa [32,33]. In this study, although the deletion of Dpi TCS was detrimental to the long-term colonization of the small intestine, the TCSs of the CitB family, including Dpi TCS, were not necessarily linked to the expression of the major subunits of the fimbrial operons (Figure 5E). We could reasonably guess there were other virulence factors influencing colonization, such as lipopolysaccharide or O-antigen capsules, which are also crucial to developing a biofilm as extracellular polymeric substances [48]. It is likely that the Dpi system influences other operons than citrate utilization, which needs more research to explore the function of the Dpi TCS regulon. 

In addition, another interesting point was that the triple-deficient strain’s role in biofilm formation, tolerance to environmental stresses and resistance to antimicrobials, and growth in egg albumen, were always more minor or similar to that of the single-deficient strains. This probably involves genetic compensation, which often happens in organisms [49]. Once a gene is lost, another will perform familiar functions and expression patterns, and thereby the organisms are capable of maintaining homeostasis. This was why we did not use the triple-deficient strain when carrying out chicken embryo and chicken experiments concerning animal welfare. Certainly, whether there is genetic compensation and which genes play a compensatory role require in-depth research. In addition, one of the limitations in this research was that we took the two genes within one TCS as a whole when exploring their functions instead studying each gene individually. Collectively, the roles and characteristics of the CitB family were elucidated in this study, which gives a better understanding of the pathogenic role of *S.* Pullorum.

## 4. Materials and Methods

### 4.1. Bacterial Strains and Growth Conditions

The bacterial strains, plasmids and primers used in this study are listed in Appendix A. The hypervirulent WT strain R51 was isolated from the liver of a sick chicken. WT, derivative mutants, *Pseudomonas*
*aeruginosa* (*P.*
*aeruginosa*) ATCC 27853, *Escherichia coli* (*E. coli*) ATCC 25922, *E. coli* TG1 and Rosetta (DE3) were grown in LB (Lennox broth) (1% tryptone, 1%NaCl, 0.5% yeast) and other culture conditions were described in the corresponding sections below. Strains with temperature-sensitive and L-arabinose-inducible expression plasmid pCas were grown at 30 °C. Arabinose (10 mM) was added to induce the expression of recombinase genes *exo*, *bet* and *gam*. Selective antibiotics were added at the following concentrations unless otherwise specified: chloramphenicol 25 µg/mL, spectinomycin 100 µg/mL and kanamycin 50 µg/mL.

### 4.2. Construction of the ΔcitAB, ΔdcuSR, ΔdpiBA and Δ3 Mutants

Four gene deletion strains were constructed using the CRISPR/Cas9-mediated genome-editing system [28]. Briefly, taking the Δ*cit* (*citAB*) mutant as an example, the sgRNA (20-bp region) was designed through the online web server (http://chopchop.cbu.uib.no, accessed on 20 May 2019) and amplified from plasmid pTargetF using the primers cit-sgRNA-F/sgRNA-R. The donor DNA fragments, 500-bp sequences homologous to each side (upstream Xf and downstream Xr) of the targeted region, shown in Appendix A in the genome, were amplified from R51 using the primer pairs cit-Donor DNA Xf-F/R and cit-Donor DNA Xr-F/R. Next, the sgRNA and Donor DNA fragments were ligated into pTargetF by overlap PCR using recombinase (Vazyme, Nanjing, China). The product, also called pTargetT, consisting of sgRNA and Donor DNA, was then chemically transformed into R51, into which the L-arabinose-induced plasmid pCas had been transformed, to generate the *citAB* deletion mutant followed by both PCR and sequencing. To cure pTargetT, the target colony harboring both pCas and pTargetT was inoculated into LB medium containing kanamycin (50 µg/mL) and IPTG (isopropyl-D-thiogalactopyranoside, 0.5 mM), and pCas was cured by growing the colonies overnight at 42 °C nonselectively. The Δ*dcuSR*, Δ*dpiBA* and Δ3 mutants were also obtained using the same method.

### 4.3. Growth Curves Measurement

Growth curves of the strains were determined by measuring the OD_620nm_ (optical density) at 2-h intervals for 12 h under different conditions. For LB shaking, overnight cultures of bacterial strains were inoculated into LB medium at 1:100 dilution and incubated at 37 °C with shaking at 180 rpm. For LB static culture, the overnight culture prepared in LB broth was diluted at 1:100 with fresh LB broth, of which 200 μL was added to a 96-well plate. Twelve plates (six for aerobic conditions and the rest for anaerobic conditions) were prepared at 6-time point and cultured in an incubator and an anaerobic jar with anaerobic paper sachets for aerobic and anaerobic growth conditions, respectively. The plates were taken out and mixed fully, and the OD value was measured at each time point. For M9 minimal medium static culture, the overnight culture prepared in LB broth was diluted at 1:100 with M9 minimal medium, of which 200 μL was added to a 96-well plate. The rest of the steps were the same as the above LB static culture conditions. One liter of M9 minimal medium consisted of 20 mM glucose (20 mL), 1 mM CaCl_2_ (0.1 mL), 1 mM MgSO_4_ (2 mL), 5× M9 salts (200 mL) and ddH_2_O. The ingredients of the 5× M9 salts were 12.8 g Na_2_HPO_4_·7H_2_O, 3 g KH_2_PO_4_, 1 g NH_4_Cl, 0.5 g NaCl and 200 mL ddH_2_O. The glucose in the M9 minimal medium was replaced with an equal concentration of citrate to investigate whether the strains utilized the citrate (Aladdin, Shanghai, China).

### 4.4. Biochemical Tests

Biochemical tests based on fifteen kinds of commercial microtubes (Land bridge, Beijing, China) were performed. According to the instructions, the OD_620nm_ of overnight cultures were adjusted to 0.4, and then a drop of the culture of about 50 μL was added to each burette. The mixture was cultivated in a 37 °C incubator for 18–24 h, and the colour change and gas production were observed. Here, bromothymol blue solution was added as a pH indicator of sugars and sugar alcohol fermentation due to acid production.

### 4.5. Biofilm Formation Assay 

The biofilm formation assay was completed based on previous experience, with minor changes [50]. Overnight cultures were adjusted to OD_620nm_ of 0.4, and 200 μL was subsequently transferred into a 96-well polystyrene microtiter plate after being diluted in a ratio of 1:10. To prevent marginalization, the same volume of sterilized ddH_2_O was added to the peripheral wells. *P.*
*aeruginosa* ATCC 27853 and LB broth, positive and negative control, were also added. Six parallel wells were settled. The 96-well plates were cultured at 20 °C, 28 °C, 37 °C and 42 °C under aerobic and anaerobic conditions. Then, after 48 h, the supernatants were discarded, followed by the plates being washed three to five times with ddH_2_O and dried in an oven. Afterwards, a volume of 200 µL of 0.4% (wt%) crystal violet was added to each well for 25 min at room temperature, and 200 μL of a 3:1 mixture of ethanol and acetone was added to elute the staining solution for about 20 min in an oscillator. Finally, the absorbance value of OD_620nm_ was measured. 

We also tested the effect of different bile salts on biofilm formation, choosing LB with 0.1% bovine bile salt (Bioway, Shanghai, China), 0.1% No.3 bile salt (Bioway, Shanghai, China) and 0.5% pig bile salt (Hongrun Baoshun, Beijing, China) as biofilm growth media separately, and allowing to the formation of biofilm for 24 h at 42 °C under aerobic and anaerobic conditions. All other steps were the same as above.

### 4.6. Antimicrobial Susceptibility Test

The broth microdilution method, as described previously [51], was used to determine the susceptibilities of the following 16 antimicrobials: gentamicin (GEN), kanamycin (KAN), streptomycin (STR), cefoxitin (FOX), ceftiofur (CF), ceftriaxone (CRO), ampicillin (AMP), chloramphenicol (CHL), ciprofloxacin (CIP), nalidixic acid (NAL), tetracycline (TET), azithromycin (AZM), imipenem (IPM), colistin (CST), amoxicillin-clavulanate potassium (AMC) and trimethoprim-sulfamethoxazole (SXT). *P.*
*aeruginosa* ATCC 27853 and *E.*
*coli* ATCC 25922 were used as quality controls. The minimum inhibitory concentration (MIC) was judged according to the standard recommended by the CLSI [29].

### 4.7. Stress Assays

Four stress experiments were carried out to compare the tolerance of five strains. The bacterial suspensions were adjusted to OD_620nm_ of 0.5, equal to 1 × 10^9^ CFU/mL (colony forming units) with sterile 10 mM PBS (Phosphate-Buffered Saline), followed by diluting to 1 × 10^6^ CFU/mL with pre-prepared 10% glucose solution for the hyperosmotic stress challenge, to 1 × 10^6^ CFU/mL with 0.9% (wt%) NaCl solution whose pH had been modulated to 3.5 for the acid stress challenge, and to 1 × 10^6^ CFU/mL with 4 mM hydrogen peroxide solution (H_2_O_2_) for the oxidative stress challenge, and then shaken at 37 °C for 30 min. To assess heat tolerance of the different mutants, the bacterial cells (10^6^ CFU/mL) were exposed to 50 °C for 10 min. Bacterial counts before and after stress were individually determined by culturing serial 10-fold dilution on LB agar plates. The survival ratio (%) was calculated as (CFU after stress/initial CFU) × 100. 

### 4.8. Growth in Egg Albumen

The ability of all strains to survive in egg albumen was quantified as described by Shah et al. [52]. Specific-pathogen-free (SPF), organic unfertilized eggs within nine days of laying (stored at 4 °C for less than a week) were decontaminated by immersing in 75% ethanol and opened from an air chamber. The egg albumen was further transferred to a sterile 50 mL centrifuge tube (about 30 mL per egg), followed by thorough mixing. A total of 100 μL of bacterial culture (bacterial concentration was equal to 2~4 × 10^3^ CFU/mL), resuspended with 10 mM sterilized PBS, was added to 900 μL of egg white in 2 mL EP centrifuge tubes before sufficient mixing. The bacterium-albumen mixture was diluted in PBS and plated on LB agar to enumerate the viable bacteria counts after incubating at 37 °C for 24 h. The survival ratio was calculated using the following formula: CFU at 24 h/CFU at 0 h.

### 4.9. Chicken Embryo Infection Model

The SPF chicken embryo model [53] was established to evaluate the virulence differences among five strains. Ninety 10-day-old SPF chicken embryos were purchased, randomly divided into five groups (*n* = 16) and inoculated with 100 μL of bacterial culture (about 50 CFU) at the 16th day of age. The chicken embryos were observed with an egg illuminator and marked as alive or dead according to the venous system’s integrity and embryo movement. Four embryos were dissected in each group until the chicks were born after five days, whose liver and spleen samples were aseptically collected and added to 500 μL of PBS for grinding for bacterial load counts.

### 4.10. Chicken Infection Model

The hatching method until 200 chicks came out of the shell at 21 days was the same as the above chicken embryo experiment without the other treatment. The experiment was designed with 40 chickens per group, 10 of which were used for observation of mortality and the rest for anatomical sampling. Every chick was orally challenged with a dose of 10^6^ CFU (200 μL) on the first day after hatching, and 200 μL of PBS was inoculated as a control. Death was observed and recorded daily during an experimental period of 21 days. On the 3rd, 7th and 14th day after the administration, three chicks were dissected in each group, respectively, and the liver, spleen, small intestine and faecal samples were collected sterilely for bacterial counting.

### 4.11. Cellular Uptake and Proliferation Assay

Chicken macrophage cell line HD-11 was cultured in RPMI 1640 medium (Thermo Fisher, Shanghai, China) supplemented with 10% fetal bovine serum (FBS) [54]. The cells were seeded in 24-well plates to a density of 2 × 10^5^ cells per well and incubated for 24 h at 37 °C in a humidified 5% CO_2_ incubator prior to infection. The overnight cultured bacteria, adjusted to an OD_620nm_ of 0.4, were washed three times and resuspended with PBS, and added to the wells at a multiplicity of infection (MOI) of 50:1 followed by centrifugation of plates at 600 g for 10 min to synchronize phagocytosis. After 1.5 h of incubation, the non-internalized bacteria were removed by washing three times with PBS and killed by cell medium containing 100 μg/mL gentamicin during another 1 h incubation. The macrophage cells were lysed, and bacteria were enumerated following serial dilution and plating onto LB agar, which was set as the 0 h time point. For the bacterial proliferation assay, the cell medium containing a high concentration of gentamicin was replaced by RPMI 1640 medium containing 20 μg/mL gentamicin, and the cell plates were incubated back at 37 °C. At 2 h, 6 h and 20 h time points, the cells were washed and bacteria were lysed to determine the number of CFU as mentioned above. Uptake rate was calculated as follows: intracellular CFU at 1.5 h/inoculum CFU, and proliferation rate was calculated as follows: intracellular CFU at time point/intracellular CFU at 1.5 h.

### 4.12. Cloning, Expression, and Purification of Recombinant Fimbrial Protein

As Liang et al. reported [55], the segment of the encoding major fimbrial subunit (*bcfA* used as an example) was amplified by PCR from strain R51 using the primers *bcfA* segment-F/R and ligated into the vector pET30a using recombinase. The ligation product was transformed into competent *E. coli* DH5α by 42 °C heat shock for 90 s. The colony harboring recombinant vector, grown on LB agar plate with 50 μg/mL kanamycin and containing the segment as determined by sequencing, was extracted and transformed into Rosetta (DE3) competent cells. IPTG (final concentration 1 mM) at an OD_620nm_ of 0.4 was added to induce recombinant protein expression at 16 °C overnight. Then, the expressed recombinant protein was purified using His-tag purification resins, whose concentration was measured by Bradford assay [56] and stored at −80 °C for later use. The other five fimbrial proteins (FimA, LpfA, SafA, StdA and SthA) were also prepared with the same strategy.

### 4.13. Rabbit Immunization and Polyclonal Antibodies Production

New Zealand rabbits weighing 1.8–2.3 kg (1 animal for each protein) were immunized by subcutaneously injecting about 1 mg of purified recombinant fimbrial proteins, which were emulsified in an equal volume of Freund’s complete adjuvant (FCA) at several locations on the back, followed by boosting immunizations every two weeks with 500 μg of protein emulsified in Freund’s incomplete adjuvant (FIA). Afterwards, blood samples were collected by arteria carotid bleeding one week after the final injection, and the serum was separated by centrifugation and stored at − 20 °C. Production of polyclonal antibodies was evaluated by indirect enzyme-linked immunosorbent assay (ELISA) against the recombinant antigens. Immunization was conducted three or four times depending on the antibody titer. One rabbit was left unimmunized, whose serum was used as a negative control. After the completion of the experiments, all of these animals were euthanized.

### 4.14. The Titer Detection of Polyclonal Antibodies

Indirect ELISA for the detection of antibody titer was established [57]. A volume of 100 µL sodium carbonate–bicarbonate buffer (1.59 g/L Na_2_CO_3_, 2.93 g/L NaHCO_3_, 0.05 M, pH 9.6) supplied with 2 μg of recombinant protein was used to coat each well of 96-well polystyrene microtiter ELISA plates overnight at 4 °C. The plates were washed three times with washing buffer PBST (Phosphate Buffered Solution with Tween-20, 500 μL Tween 20 in 1 L 10 mM PBS, pH 7.4) and blocked for 2 h at 37 °C with 200 μL blocking buffer (5% skimmed milk in PBST) followed by three washes. Subsequently, serial 2-fold dilution of rabbit serum ranging from 1:5000 to 1:640,000 was applied in blocking buffer and added for 1 h at 37 °C. After three washes, 100 μL of goat anti-rabbit antibody conjugated to HRP at 1:5000 in blocking buffer was added to incubate for 1 h at 37 °C, followed by three washes. A total of 100 μL 3,3′,5,5′-tetramethylbenzidine (TMB) was applied. Then, the reaction was stopped after 10 min in the dark with the addition of 50 μL 1 M H_2_SO_4_. Finally, a plate reader measured the absorbance at 450 nm immediately.

### 4.15. Western Blotting Analysis

Based on previous research [58], the protein samples of whole bacteria (WT and 4 mutants) were prepared by static culture for about 36 h, under which the fimbrial protein was better expressed, resuspended and adjusted to OD_620nm_ of 0.4. Subsequently, they were added with a 5× sodium dodecyl sulfate–polyacrylamide gel electrophoresis (SDS-PAGE) sample loading buffer followed by 10 min at 100 °C. Then, equal volumes of proteins were loaded onto a 12% SDS–PAGE gel (Fude, Hangzhou, China) for 1 h at 120 V and transferred to the polyvinylidene fluoride (PVDF) membrane, which was blocked in 5% skimmed milk in TBST (Tris-HCl Buffered Saline with Tween-20, 8.75 g NaCl, 2.42 g Tris, 500 μL Tween 20, 750 μL HCl in 1 L ddH_2_O, pH 7.4) for 45 min at 37 °C followed by three washes with TBST. Then, the membranes were incubated at 4 °C with the above fimbrial antibodies and GAPDH antibody with a dilution of 1:1000 overnight. After three washes, a 1:5000-diluted goat anti-rabbit IgG secondary antibody was used for soaking the membrane for 2 h at room temperature. Finally, the protein expression was visualized using an ECL (enhanced chemiluminescence) Plus system after five washes with TBST [55].

### 4.16. Ethical Statement

All animal experiments in this research were undertaken with the permission of the Animal Experimental Ethics Committee of Zhejiang University (ZJU) and the approval document (ZJU20190094).

### 4.17. Statistical Analysis

Unless otherwise specified, all values are means from three separate experiments ± standard deviation. Statistical differences between the results were analyzed using one-way analysis of variance (ANOVA) using the GraphPad Prism software version 9.0. A value of *p* < 0.05 was considered statistically different.

## Figures and Tables

**Figure 1 ijms-23-10201-f001:**
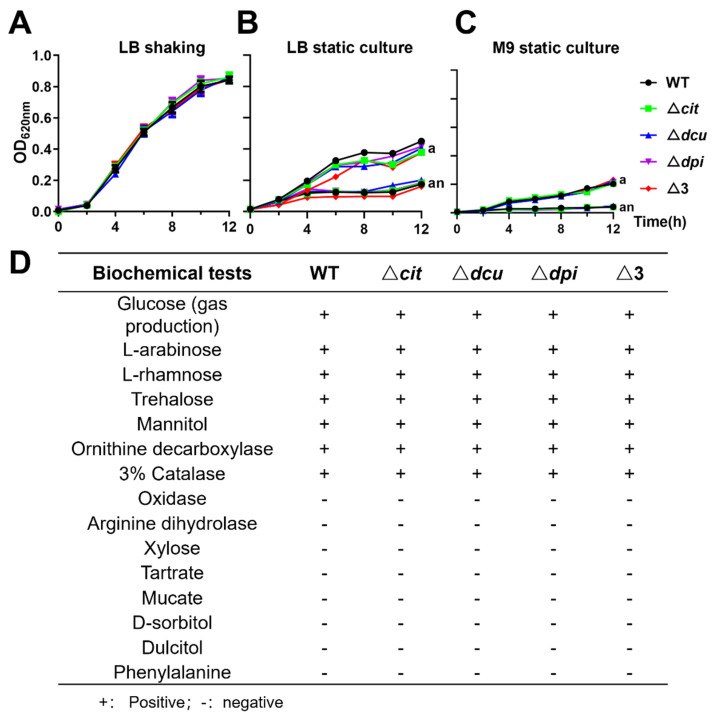
Deletion of TCSs of the CitB family did not affect the growth and the results of biochemical tests. (**A**–**C**) Growth curves of WT R51 and mutants under LB shaking conditions (**A**), LB static culture (**B**) and M9 minimal medium static culture (**C**) (a, aerobic conditions; an, anaerobic conditions). (**D**) The biochemical test results of WT and mutants.

**Figure 2 ijms-23-10201-f002:**
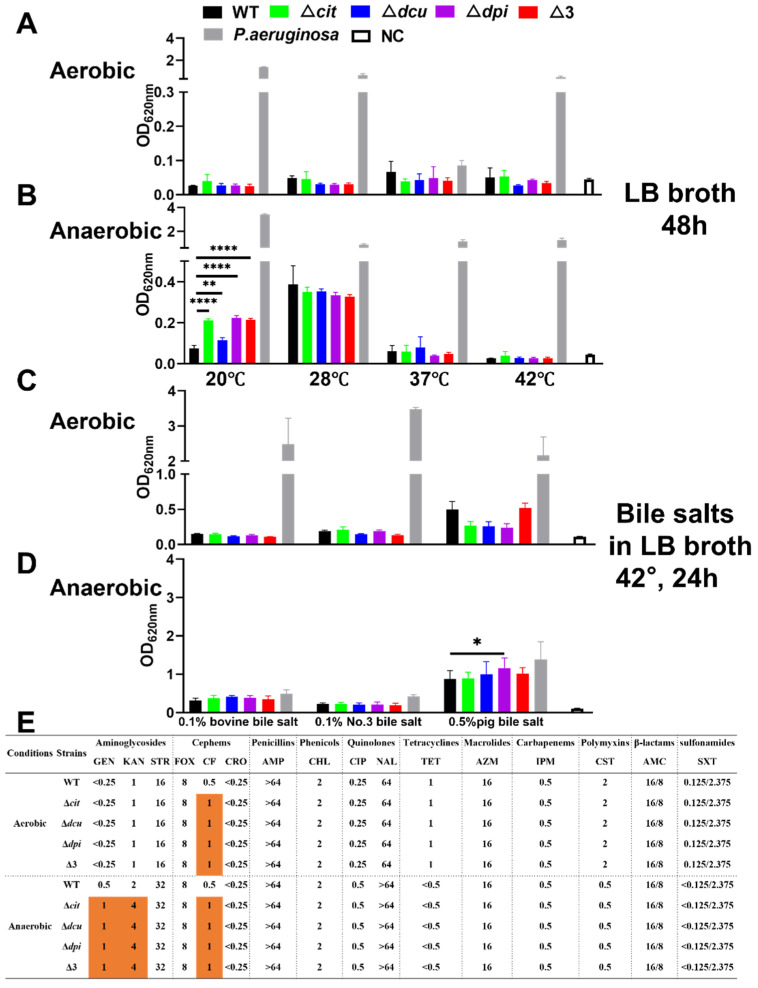
Deletion of TCSs of the CitB family increased the biofilm formation and the antibacterial resistance to aminoglycoside drugs under anaerobic conditions. (**A**,**B**) Determination of biofilm formed by WT and mutants at 20 °C, 28 °C, 37 °C and 42 °C cultured in LB broth for 24 h under aerobic conditions (**A**) and anaerobic conditions (**B**). (**C**,**D**) Biofilm formation in different bile salts at 42 °C for 24 h under aerobic conditions (**C**) and anaerobic conditions (**D**). (**E**) MIC values of WT and mutants against the tested antimicrobial agents. The abbreviations of antimicrobial agents are listed in Section 4.6. Orange-filled numbers indicate that the MIC values of the mutants are greater than those of the WT. Statistical significance of differences was evaluated by one-way ANOVA test (* *p* < 0.05; ** *p* < 0.01, **** *p* < 0.0001).

**Figure 3 ijms-23-10201-f003:**
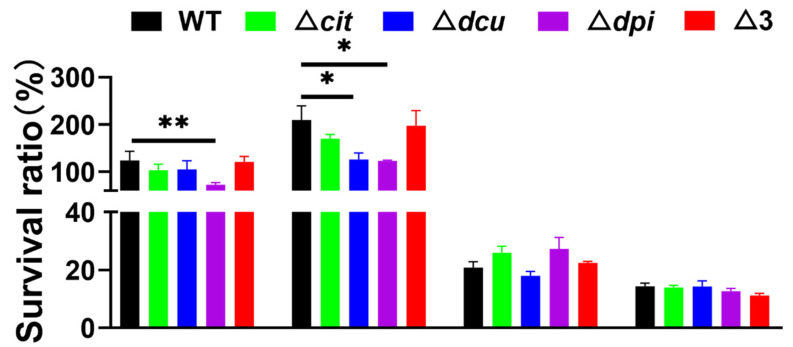
Hyperosmotic, oxidative, acid and heat stress assays. Treatment conditions from left to right were 10% glucose solution for 30 min, 4 mM hydrogen peroxide solution for 30 min, NaCl solution with pH 3.5 for 30 min and 50 °C for 10 min, respectively. Statistical significance of differences was evaluated by one-way ANOVA test (* *p* < 0.05; ** *p* < 0.01).

**Figure 4 ijms-23-10201-f004:**
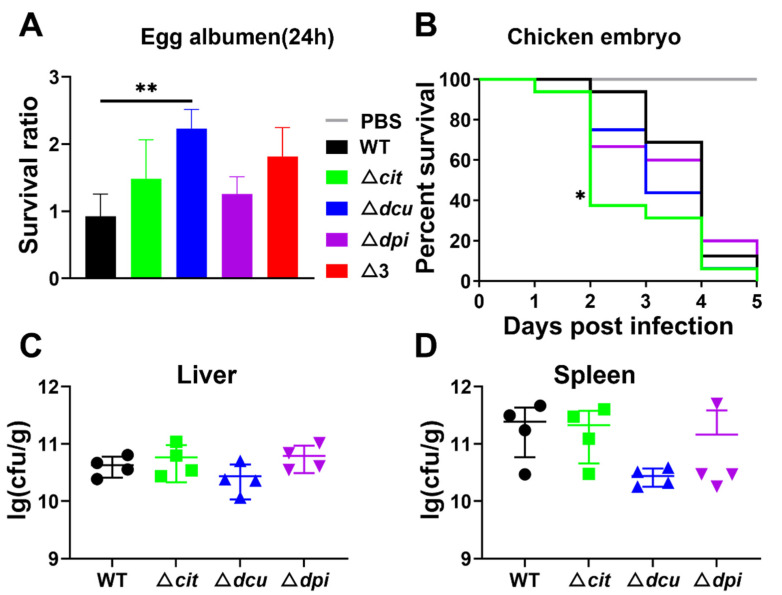
Deletion of TCSs of the CitB family increased the growth in egg albumen and virulence of the chicken embryo. (**A**) The survival of WT and mutants after 24 h cultured in egg white at 37 °C and the ratio calculated as CFU (colony forming units) at 24 h/CFU at 0 h. (**B**) Survival curves for chicken embryos infected with WT and mutants. Sixteen-day-old chicken embryos were inoculated, each strain with 50 CFU via allantoic cavity injection and monitored for five days until hatching out of the shell. (**C**,**D**) Bacterial loads in liver (**C**) and spleen (**D**) tissues of infected embryos 5 days post-infection. Statistical significance of differences was evaluated by one-way ANOVA test (* *p* < 0.05; ** *p* < 0.01).

**Figure 5 ijms-23-10201-f005:**
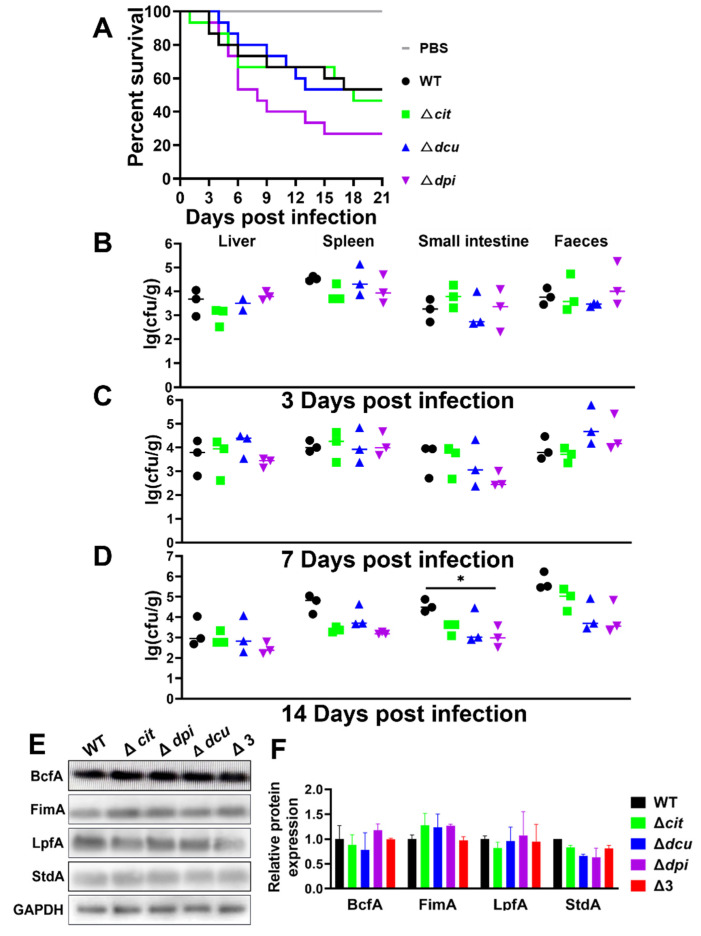
Deletion of Dpi TCS was detrimental to long-term colonization of the small intestine but did not affect the expression of the fimbrial protein. (**A**) Survival curves for chickens infected with WT and mutants. One-day-old chickens were inoculated, each strain with 10^6^ CFU, by oral injection and monitored for 21 days. (**B**–**D**) Bacterial loads in the liver, spleen, small intestine tissues and faecal samples 3 (**B**), 7 (**C**) and 14 (**D**) days post-infection. (**E**) Western blotting analysis detected fimbrial protein expression of WT and mutants. The GAPDH was used as a reference. (**F**) The relative expression level of four fimbrial proteins in WT and mutants. The statistical significance of differences was evaluated by one-way ANOVA test (* *p* < 0.05).

## Data Availability

The data presented in this study are available in Appendix A.

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
