# Peer review of "Characterization of Two-Component System CitB Family in Salmonella Pullorum"

_ijms, 2022, doi:10.3390/ijms231710201_

Round 1
Reviewer 1 Report
Reviewer's Comments
Title: Characterization of Two-component System CitB family in Salmonella Pullorum
MS ID: ijms-1881983
In the present, the authors investigated the CitB family in S.pullorum. and concluded that it does not affect growth and metabolism. And provided a new function for CitB in virulence.
1. Abstract well written and composed of understandable results and conclusions.
2. Introduction is very precise and straight with the topic.
3. Methods used are very advanced, mutant strains were constructed using an advanced technique, CRSPR/Cas9 editing system.
4. In results Fig. 1 and table 1 provide knowledge on the effect of the CitB family on growth and metabolism, sugar utilization.
5. Extensive discussion is supporting the resutls.
Reviewer 2 Report
Overview comments
The submission explores the role of one of the two component systems (TCS) present in a chicken pathogen, Salmonella Pulloram, the CitB family which contains three TCSs. The authors developed knockout mutants of each of these three systems and describe some of the physiological properties (utilisation of selected carbon sources, other biochemical tests and growth rates under aerobic and static culture conditions; antibiotic resistance; survival after exposure to stressors) and traits potentially associated with pathogenicity (production of fimbriae proteins; survival in chicken embryos; clearance/infectivity in chickens). The submission represents a considerable body of work and the authors should be congratulated on this initiative. There are some issues with the manuscript which require addressing: some are editorial/phraseological, while others are more substantive matters related to presentation of data, its interpretation and its limitations. Some of these matters may be clarified when the Materials and Methods are filled out/corrected and the various comments made on the annotated pdf are addressed. Specific comments follow below.
Materials and Methods
This section requires some revision so that the exact methods used are clearer; many of the issues raised are marked on the pdf provided and include:
· The phrasing in several of the sections requires clarity (sentence construction is sometimes confused and the exact methods used are not understandable)
· Many sentences start with numbers: check journal format and correct as required (many instances noted throughout the text)
· Many of the methods do not have citations
· Some abbreviations are not spelt out – many are obvious while others are not
· The suppliers of some of the materials are not provided, specifically the biochemical tests used. Check journal format for the requirement to provide supplier and location and fix all instances
· M9 is mentioned in the Results, but there is no in the Methods section describing media or reference to composition
· The Pseudomonas strain appearing in Fig. 2 is not mentioned in 4.1, other strains also missing? Consolidate similar items under the appropriate sections.
Some of the above are significant for interpreting the results and need addressing. All of the ‘comment’ bubbles on the pdf require consideration by the authors and rectification.
Results
There are numerous ‘comment bubbles’ provided on the pdf, concerning how data is presented in the figures (using abbreviations that are the same as one of the mutants; using A and B in panels and in other annotation on the diagrams – simply confusing) and the interpretation of data (for example, some of the antibiotic MIC differences between mutant and WT are not convincing, as they may represent a single tube difference in MIC; Western Blot data was qualitative, presented as an image, not quantitative – and this could have been quantified to show subtle differences in protein levels against the control, GAPDH and some of the densities may have indicated that expression levels differ but the text says otherwise).
The authors need to address each one of the queries to provide greater clarity and to recognise the limitation of the data and the conclusions drawn. Review the legends of figures to make sure that abbreviations are obvious and refer to the appropriate paragraph in section 4 as needed to define the abbreviations.
Discussion
Lines 192-195: there are two mentions of ‘data not shown’. One of these is for growth in LB or M9 with citrate. Given that one would anticipate that if the CitB TCSs are involved in citrate metabolism (uptake and use as a C/energy source), then knockout mutants, especially for the triple mutant, would have some impact on growth on citrate. The data for growth on citrate should appear somewhere in the manuscript, even if this is in ‘supplementary materials’ or a statement regarding results being similar (for example) as growth on glucose (if this is the case) made clearly in the text. This is a critical point, as the inference is that other regulatory networks are involved in citrate utilisation in addition to the three TCSs examined here, if all of the mutants can still grow on citrate. The sentences in this paragraph seem to be contradictory – be clear about whether the mutants can grow on citrate or not. If they can grow on citrate still, it may suggest that the promoter regions of the regulons impacted by the CitB family of TCSs are also regulated by other regulators – which happens in many systems particularly where global regulators play a role in gene expression. The authors need to consider this as part of the discussion, rather than just referring to genetic compensation later (line 233). It is likely that the CitB family of regulators have a broader regulon, including influencing cell surface component synthesis and intermediary metabolism – to be explored in the future, no doubt, but what is known in other Gram negatives and other phyla?
The authors need to revise the Discussion in context of recognising the limitations of the data and where the data are unequivocal. Sorting out the broader regulon of the CitB-type TCSs and, conversely, the other regulators what may influence citrate utilisation will clearly be the subject for further research, although the current submission provides a good starting point in generating a family of mutants and defining some of the physiological properties of these mutants in context of pathogenicity regulation.
Recommendation
The submission requires considerable revision and should be resubmitted for further review.

Reviewer 3 Report
This article is very interesting that a number of various experiments have been conducted to investigate the roles of CitB family Sa. Pullorum. The researchers have made a lot of efforts to decipher the difference between WT and mutant strains. Beyond the methods, the data are analyzed in scientifically sound ways and results presented in clear and understandable ways. A little concern a have is that the main findings are not discussed enough in the light of what has been known so far (compared to other studies).
S figure 2 – a legend is required
I would suggest providing a supplementary material that describes the mutant creation for the four mutants.
It looks having citB family incurs costs to the Salmonella. For instance, the mutants are tolerant to hyperosmotic and oxidative stresses, meaning the presence of citB genes decrease its survival in harsh environments. If this gene family works again Salmonella organisms, why do they keep it? its deletion results in increased resistance to antibiotics; conversely, its presence renders Salmonella susceptible to antibiotics.

Round 2
Reviewer 2 Report
The authors have address all of the matters raised in the review report and in the annotated pdf file. Well done to them. A few minor errors have crept into the text during revision e.g. line 414, should be carotid (not carotid); one sentence now starts with And. However, these very minor issues can be dealt with during the editorial process.
For the authors’ information, in other species the Cit operon can have several deletions so that growth on this substrate may be impaired. It would be worth looking at this region for presence and evolution, to account for lack of growth on citrate (which sometimes only occurs when catabolite repression ceases late in the growth cycle when another carbohydrate source is present). The regulon related to citrate use as a sole C source in other species includes induction of genes associated with oxidative stress, so it is not too surprising that deletion of the TCS had an ‘unanticipated’ impact.
Recommendation: accept, and pick up the minor typographical matters during processing for publication.